# Microplastic detection and remediation through efficient interfacial solar evaporation for immaculate water production

Zhen Yu[1,2], Yang Li[2], Yaoxin Zhang[3], Ping Xu [4], Chade Lv [4], Wulong Li[5], Bushra Maryam [2], Xianhua Liu [2]✉ & Swee Ching Tan [1]✉

Freshwater scarcity and microplastics (MPs) pollution are two concerning and intertwined global challenges. In this work, we propose a "one stone kills two birds" strategy by employing an interfacial solar evaporation platform (ISEP) combined with a MPs adsorbent. This strategy aims to produce clean water and simultaneously enhance MPs removal. Unlike traditional predecessors, our ISEP generates condensed water free from MPs contamination. Additionally, the photothermally driven interfacial separation process significantly improves the MPs removal performance. We observed a removal ratio increase of up to 5.5 times compared to previously reported MPs adsorbents. Thus, our rationally-designed ISEP holds promising potential to not only mitigate the existing water scarcity issue but also remediate MPs pollution in natural water environments.

Plastic pollution is undoubtedly one of the most pressing global challenges we face today[1]. A staggering amount of plastic has been produced in recent decades, but only 26% is recovered for recycling[2]. The remaining unrecycled plastic fragments disintegrate into microscopic particles, known as MPs, due to natural decompositions such as sunlight, air, oceanic forces, and biological processes[3,4]. MPs are pervasive in water, soil, and air, and can be extracted by using simple filtration-digestion methods (Supplementary Fig. 1)[5,6]. While pristine filter membranes are free of MPs (Supplementary Fig. 2), numerous MPs particles or fibers are observed on the filter membranes after filtering digestion solutions sampled from rivers, soil, and air (Fig. 1a). To exclude the effect of the regional climate variations, we collected water samples from Singapore (tropical) and the Tibetan Plateau (high altitude, known as the roof of the world). MPs were detected in these samples as well (Fig. 1a), indicating that MPs are widely distributed in nature. More importantly; it's not just

the pure MPs, a large number of other pollutants from the surroundings usually bond with MPs. The combined toxicity of these pollutants and MPs poses a significant threat to living organisms[7–9].

In recent years, research efforts have predominantly focused on upcycling processes and eco-environmental treatments for plastic fragments[10]. Advanced technologies, such as microbial and catalytic methods, have yielded exciting results[11–15]. However, these methods cannot directly address MPs in natural environment unless pre-recycling, given the low concentration of MPs and complex practical conditions[16]. Some physical technologies have emerged as effective solutions for treating MPs[17]. For instance, Fang et al. recovered MPs through microbubbles produced by focusing sunlight through a glass ball[18]. While this technology offers lower energy consumption and operational costs compared to traditional air flotation technology, the electrostatic repulsion between microbubbles and negatively charged

[1]Department of Materials Science and Engineering, National University of Singapore, Singapore, Republic of Singapore. [2]School of Environmental Science and Engineering, Tianjin University, Tianjin, P. R. China. [3]China-UK Low Carbon College, Shanghai Jiao Tong University, Shanghai, P. R. China. [4]MIIT Key Laboratory of Critical Materials Technology for New Energy Conversion and Storage, School of Chemistry and Chemical Engineering, Harbin Institute of Technology, Harbin, P. R. China. [5]School of Electrical and Electronic Engineering, Nanyang Technological University, Singapore, Republic of Singapore. ✉e-mail: lxh@tju.edu.cn; msetansc@nus.edu.sg

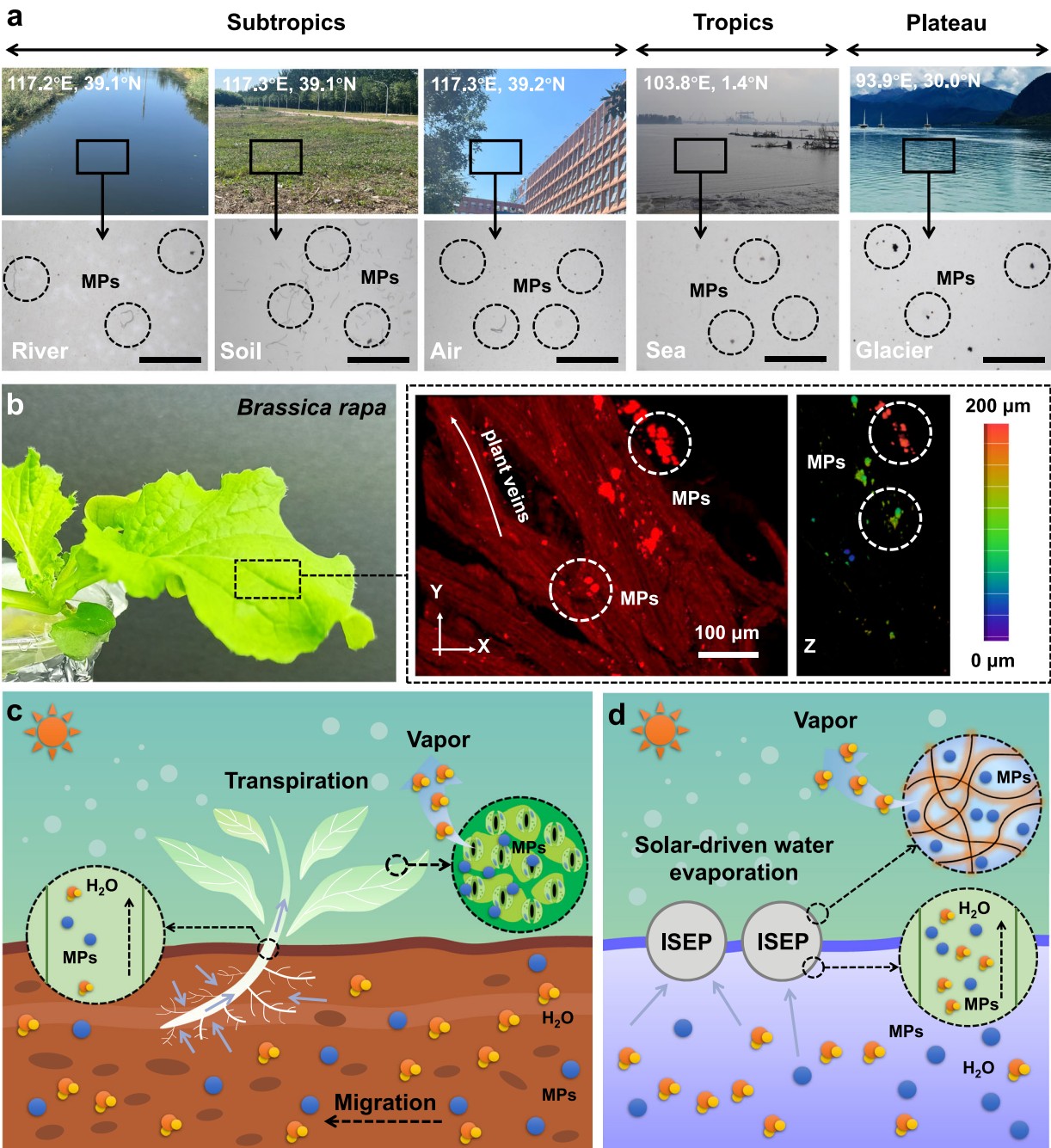

**Fig. 1 | The inspiration of the "One stone and two birds" strategy. a** Microscopic images of the filter membrane after filtering digestion solutions from different sampling points. Scale bar: 100 μm. **b** Digital image of *Brassica rapa* and Raman confocal microscopy images of its leaves. The bar is the vertical distance from the top of the leaves. **c** Schematic diagram of the migration process of MPs from the soil to plants through transpiration. **d** Schematic diagram of ISEP that removes MPs from water and simultaneously produces clean water.

MPs presents challenges for efficient collection[19]. Moreover, this approach is limited in producing clean water from real seawater or wastewater due to the presence of other pollutants or salts. Adsorption methods have demonstrated effectiveness in recovering MPs from marine environments[20]. However, such methods primarily focus on powder adsorbents, which often exhibit low adsorption rates, pose challenges in the recovery processes, and may even lead to potential secondary pollution[21]. Therefore, it is imperative to explore alternative methods that could offer more efficient and environmentally sustainable solutions for MPs recovery.

Freshwater scarcity is another significant global concern[22]. Interfacial solar evaporation technology holds promise in alleviating this

issue[23–26]. Over the past few years, substantial progress has been made in enhancing evaporation performance through materials design, interfacial engineering, and thermal management[27–32]. However, previous efforts have primarily focused on increasing the evaporation rate rather than considering water quality[33,34]. It is not sufficient to determine the suitability of water produced by interfacial solar evaporators for drinking solely by measuring the ion concentration, total dissolved solids (TDS), and total organic carbon (TOC) of the distilled water. Some new contaminants may be overlooked, such as MPs. MPs could potentially evaporate from water into the air through the transpiration process[35,36]. Given that the interfacial solar evaporation technology presents a stronger evaporation capacity over the transpiration

process, MPs seem to be enriched in distilled water. However, to date, the existence of MPs in the as-obtained distilled water was not adequately explored. This highlights a gap in our understanding and emphasizes the need for further research to investigate the potential presence of MPs in water produced through interfacial solar evaporation technology.

In nature, plants inevitably adsorb MPs from water or soil. Using *Brassica rapa* as an example, we introduced MPs into their growing water-soil environment and detected MPs in the leaves after 14 days (Fig. 1b). The entry of MPs into plants primarily occurs through the following process (Fig. 1c): 1) As plants absorb water, transpiration drives MPs to migrate towards the roots; 2) The thin epidermis at the junction between the main root and lateral roots of plants possess large pores, allowing MPs to cross the barrier and enter the xylem vessels of the roots. These MPs can then be transported to the stem and leaf via transpiration; 3) Water exits the leaves into the air through the stomata, while MPs remain in the leaves. Inspired by this natural process, we propose an innovative ISEP fabricated with MPs adsorbent (Fig. 1d), which significantly enhances MPs removal. It is noteworthy that the ISEP generated condensed water free from MPs under the effect of the MPs adsorbent, a feature absent in traditional solar

evaporation structures. Finally, we discuss the application prospects of ISEP for removing MPs from marine industrial products and in the context of earth circulation.

## Results and discussion
### Fabrication and characterization
The materials used to construct the ISEP should have the following essential characteristics:

1) Hydrophilic porous structure and stable photothermal conversion performance. Commercial carbon felt (CF) fuls such requirements[37]. 2) To achieve the removal of MPs, CF should be capable of adsorbing MPs. Hence, we deposit polyethyleneimine (PEI) in situ on CF, as PEI is known for its effective MPs absorption. CF possesses a porous structure with relatively rough carbon fibers inside (Fig. 2a) [17,38]. Upon grafting PEI onto CF, the porous structure of PEI-deposited CF (CF-PEI) remains intact, but the surface of the constituent fibers becomes exceptionally smooth (Fig. 2b). X-ray photoelectron spectroscopy (XPS) is then employed to analyze the surface element distribution. XPS analysis reveals a significant increase in nitrogen content in CF-PEI compared to pristine CF, potentially attributed to the nitrogen-containing groups of PEI (Supplementary Fig. 3). Further

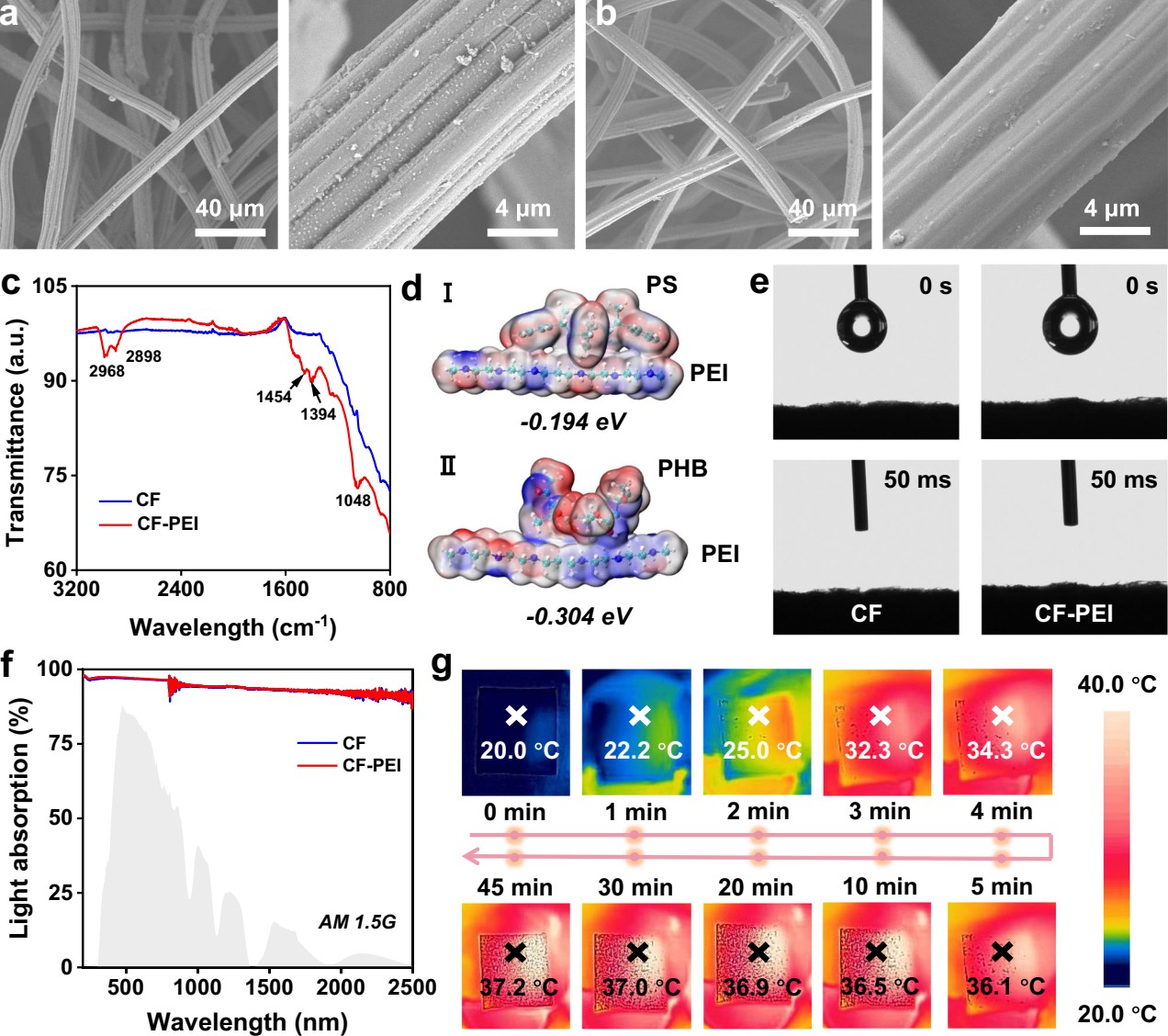

**Fig. 2 | The characterization of CF-PEI.** SEM images of (**a**) CF and (**b**). CF-PEI. **c** FT-IR spectra of CF and CF-PEI. **d** DFT calculation for the adsorption energy of PEI for PS molecules and PHB molecules. **e** The water contact angle of CF and CF-PEI. **f** The UV-Vis-NIR spectra of CF and CF-PEI. **g** The IR photos of wet CF-PEI under 1 sun.

analysis with Fourier-transform infrared spectroscopy (FT-IR) reveals novel peaks on CF-PEI compared to pristine CF, corresponding to functional groups of PEI (Fig. 2c). The peaks at about 2968 and 2898 cm$^{-1}$ are related to -CH$_3$ and -CH$_2$ groups of PEI[39,40], respectively. The peaks at 1452 and 1048 cm$^{-1}$ correspond to C-N stretching vibrations[41,42]. The peak at 1394 cm$^{-1}$ is caused by the C-H bending of alkanes. These functional groups provide numerous adsorption sites for MPs with different electronegativities[43]. Density functional theory (DFT) calculation results show that the adsorption energies of PEI for polystyrene (PS) molecules with a positive charge and poly-hydroxybutyrate (PHB) molecules with a negative charge are similar, which are both negative (Fig. 2e). As such, it can be concluded that the MPs adsorption by PEI is spontaneous and has a good adsorption effect on MPs with different electronegativities.

Compared to CF, CF-PEI also maintains commendable wetting performance (Fig. 2e). Similarly, there is little difference in the light absorption (Fig. 2f) and photothermal conversion ability between CF and CF-PEI. Under 1 sun, CF-PEI and CF in the dry state can reach a high temperature over 70 °C within 10 minutes (Supplementary Fig. 4). CF-PEI in the wet state exhibits a lower steady-state temperature, $ca$ 37 °C (Fig. 2g). We also examined the stability of CF-PEI and found little change in its surface morphology in solutions with varying pH values (Supplementary Fig. 5), demonstrating the excellent environmental tolerance of CF-PEI.

### Solar-driven clean water production

Our initial focus is to evaluate the evaporation performance of various evaporation structures, including pure water, two-dimensional ISEP (2D-ISEP), and three-dimensional ISEP (3D-ISEP). Both 2D-ISEP and 3D-ISEP are fabricated by CF-PEI unless stated otherwise. After 60 minutes, pure water exhibits a low temperature, whereas both the 2D-ISEP and 3D-ISEP show a significantly increased average temperature under 1 sun (Fig. 3a). All structures exhibit low evaporation rates under dark conditions, while the evaporation rates significantly increase under 1 sun (Fig. 3b). The evaporation rate of 3D-ISEP reaches 2.10 kg m$^{-2}$ h$^{-1}$, approximately 4.5 times and 0.6 times higher than that of pure water and 2D-ISEP, respectively. We further track the evaporation performance under low solar fluxes. 3D-ISEP still enables a stable evaporation rate of 0.61 kg m$^{-2}$ h$^{-1}$ under 0.2 sun (Supplementary Fig. 6). These results suggest that 3D-ISEP exhibits excellent evaporation performance even under weak solar irradiation, making it suitable for real-world application. For other extreme environments (such as change-able weather, low temperature, no light, etc.), introducing phase change materials or waste heat to strengthen the evaporation process is needed[44,45]. The evaporation performance of 3D-ISEP is then examined under 1 sun in various solutions with different MPs sizes, pH values, and MPs concentrations, using PS as an example. Variations in pH values, MPs concentrations, and MPs sizes have negligible effects on the evaporation performance (Supplementary Fig. 7). Unless otherwise specified, ISEP mentioned below refers to 3D-ISEP.

The evaporation performance of ISEP made of bare CF shows little difference from that of CF-PEI in MPs solution (Fig. 3c). A homemade device with a semicircular top is used to collect the condensed water produced by ISEP (Supplementary Fig. 8). Interestingly, flow cytometry spectra imply that some MPs are observed in the condensed water produced by ISEP made of CF, while the condensed water produced by ISEP made of CF-PEI is free of MPs (Fig. 3d). The excellent shielding effect of CF-PEI on MPs could be related to the adsorption effect of CF-PEI itself. To determine the adsorption limit value, we measure the water production and purification performance of ISEP in the above PS solution. The condensed water was collected every 6 h and analyzed. Over a continuous 60 h period, 3D-ISEP maintained a stable average water collection rate of 1.21 kg m$^{-2}$ h$^{-1}$ and little MPs could be detected in the condensed water (Supplementary Fig. 9). Extending the operation time to 66 h, shows little change in the average water production

rate, but some MPs appear in the condensed water, this may be related to the adsorption saturation of CF-PEI on MPs. In this state, to ensure the interception effect of MPs, it is necessary to regenerate CF-PEI, as discussed below. It is noted that the concentration of MPs in the real ocean is much lower than the above MPs solution (0.2 ~ 351 $vs$ 69687 item m$^{-3}$)[46–53]. Under this condition, it is estimated that CF-PEI needs to be regenerated only when being used for at least 496 days (The related details are provided in Supplementary Method). Given lower outdoor light intensity, regeneration time may take longer. Similar to RO membranes, this regenerate frequency is acceptable. Therefore, during the long-term operation, MPs would not block the ISEP before regeneration and deteriorate the evaporation performance.

Finally, outdoor experiments were conducted to investigate the practical performance of ISEP. The sampling points are illustrated in Supplementary Fig. 10. Large-size CF-PEI (Supplementary Fig. 11) was used to prepare ISEP for the outdoor experiments. The outdoor device was similar to the indoor device. It is important to note that ISEP does not have an active condensation module and mainly relies on gravity to collect condensed water. Therefore, when the outdoor temperature is higher than the freezing point (~ 4°C), condensed water can be obtained, which endows it with a huge practical application potential. The weather conditions during the experiment are shown in Supplementary Fig. 12a. During the initial two hours of the outdoor experiment, a significant amount of water mist formed on the top of the device and gradually accumulated at the bottom due to gravity (Supplementary Fig. 13). The device achieved a maximum water production rate of 1.62 kg m$^{-2}$ h$^{-1}$ (Supplementary Fig. 12b). The original seawater contained humic acid analogs, whereas these contaminants were not detected in the condensed water (Supplementary Fig. 14). Compared to the original seawater, the ion concentrations of the condensed water were significantly reduced, lower than the drinking water limit of WHO and EPA (Supplementary Fig. 15)[54,55]. Meanwhile, some particulate matters existed in the original seawater, but the condensed water did not contain them (Fig. 3e). The in-situ Raman confocal method was used to further determine these particulate matters. The confocal microscopic imaging results also show similar results (Fig. 3f). The in-situ Raman spectroscopy revealed intense Raman peaks at 943, 1379 and 1583 cm$^{-1}$. These are characteristics of PS and the Raman peaks (at 837, 1330, 1460 cm$^{-1}$) validating the presence of PP. Based on these Raman peaks, we can conclude that the above-mentioned particles are all MPs. To sum up, ISEP based on CF-PEI could produce MPs-free clean water.

Except for water quality, stability is another factor determining the practical potential of ISEP. The outdoor desalination process of ISEP operated in an intermittent operation mode in real scenarios[56,57]. In this mode, the salt ions accumulating in ISEP during the daytime will diffuse back into the bulk seawater at night due to the concentration differences, thereby achieving stable desalination[58–60]. To verify this, we conducted long-term solar desalination experiments in actual seawater with this mode. During the 70-hour test, ISEP maintained an evaporation rate of ~1.95 kg m$^{-2}$ h$^{-1}$ (Supplementary Fig. 16), demonstrating its stable performance in solar-powered desalination of ISEP.

### MPs removal performance and mechanism

Initially, the MPs removal abilities of ISEP based on CF and CF-PEI were compared using PS as an example. ISEP based on CF exhibits no discernible capacity to remove MPs in darkness (Fig. 4a), while ISEP based on CF-PEI demonstrates a removal efficiency of approximately 18% within 6 h. The fluorescently labeled MPs do not experience significant degradation when exposed to light (Supplementary Fig. 17). Note that ISEP achieves an enhanced MPs removal ratio of around 100% within 6 h under 1 sun (Fig. 4a), with the MPs evenly distributed on the fibers of CF-PEI in ISEP (Supplementary Fig. 18).

This superior MPs removal performance can be attributed to three key factors:

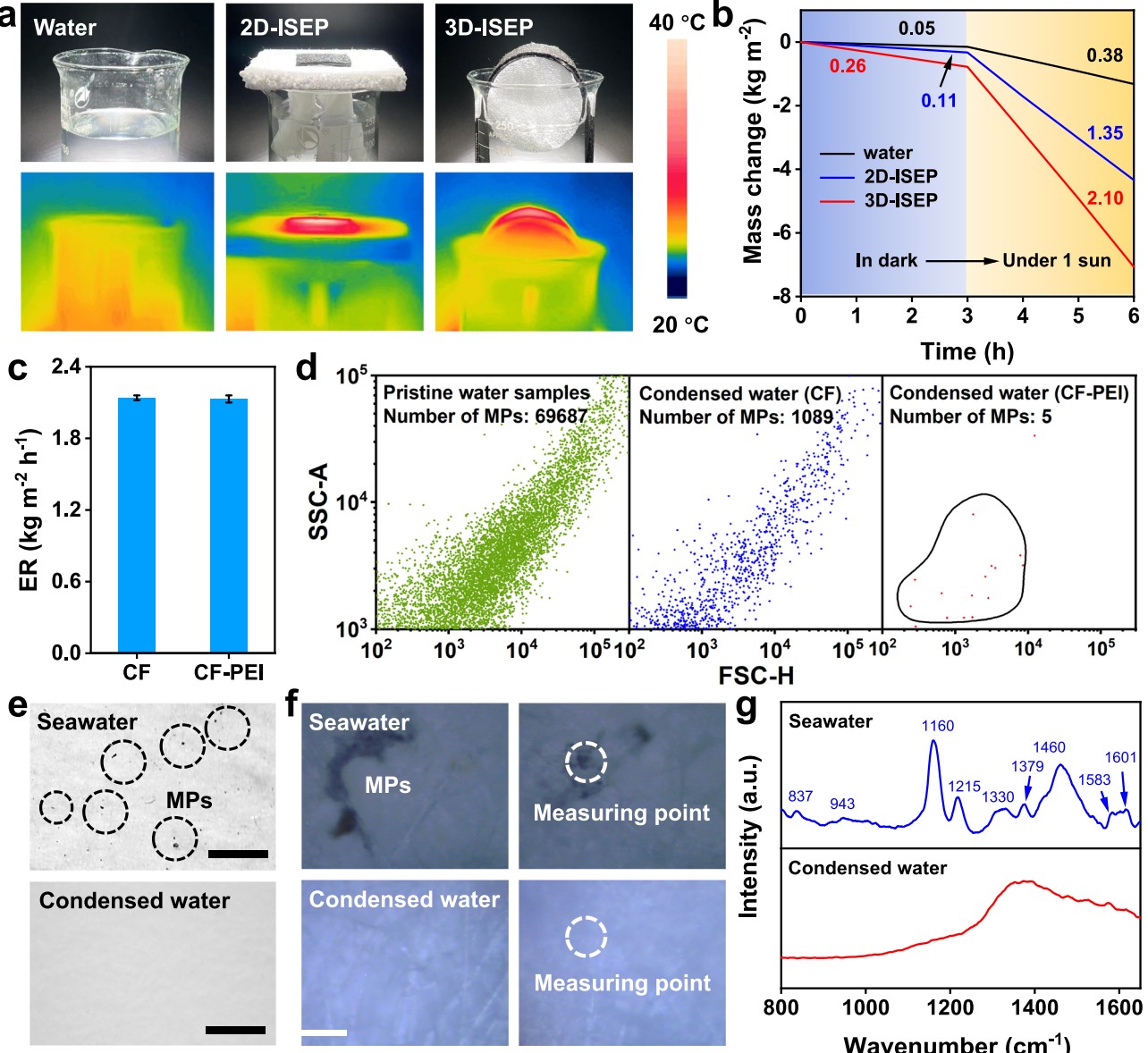

**Fig. 3 | The evaporation performance of 3D-ISEP. a** The digital photos and IR photos of pure water, 2D-ISEP, and 3D-ISEP under 1 sun. **b** The mass change curves of pure water, 2D-ISEP, and 3D-ISEP in the dark and under 1 sun. **c** The evaporation rate of 3D-ISEP based on CF or CF-PEI in 15 mg L⁻¹ PS solution. Initial conditions: PS concentration is 15 mg L⁻¹, the pH of PS solution is 7, and the particle size of PS is 0.5 - 80 μm. Error bars represent the standard deviations from three-time measurements. **d** Number of MPs (PS, mixed size, negatively charged) in different water samples determined by flow cytometry. **e** Microscopic images of the filter membrane after filtering the original seawater and condensed water. Scale bar: 100 μm. **f** The in-situ Raman confocal microscopic images of the filter membrane after filtering the original seawater and condensed water. Scale bar: 10 μm. **g** The in-situ Raman spectra of the measuring point of the filter membrane after filtering the original seawater and condensed water.

(1) The MPs absorption by CF-PEI (Fig. 4b). FT-IR measurements have been conducted to study this adsorption process (Supplementary Fig. 19). After adsorbing MPs, the peaks of CF-PEI at 2968, 2898, 1454, 1394, and 1048 cm⁻¹ all disappear, indicating that their involvement in the MPs adsorption. Further investigation through XPS measurements reveals the disappearance of C = O bonds, C-N bonds, and C = N bonds after MP adsorption (Supplementary Fig. 20a). The C-N bonds on the CF-PEI are gradually weakened (Supplementary Fig. 20b), consistent with FT-IR results, suggesting the importance of these chemical bonds in MPs adsorption. Meanwhile, a new C-N bond belonging to PS appears on the CF-PEI after adsorbing PS (Supplementary Fig. 20b). Based on the FT-IR and XPS results, MPs adsorption by CF-PEI is highly related to its functional groups. In addition, physical electrostatic adsorption also contributes to MPs adsorption, as PS is

negatively charged while CF-PEI is positively charged (Supplementary Tables 1 and 2). After CF-PEI adsorbing PS, the surface potential of CF-PEI changes from positive to negative (Supplementary Table 2). Therefore, the MPs adsorption by CF-PEI strongly relies on chemical adsorption and physical electrostatic adsorption.

(2) The photothermal enhanced effect. The surface temperature of ISEP increases significantly with solar flux rising, facilitating the endothermic adsorption process of MPs. Notably, ISEP based on CF-PEI only achieves an increased MPs removal efficiency of ~27% within 6 h at 60 °C in the dark (Supplementary Fig. 21). Thus, the photothermal effect cannot be considered as the primary driver to accelerate MPs removal.

(3) Evaporation enhanced effect (Fig. 4b). Under solar irradiation, the evaporation process of ISEP can facilitate the transmission of MPs,

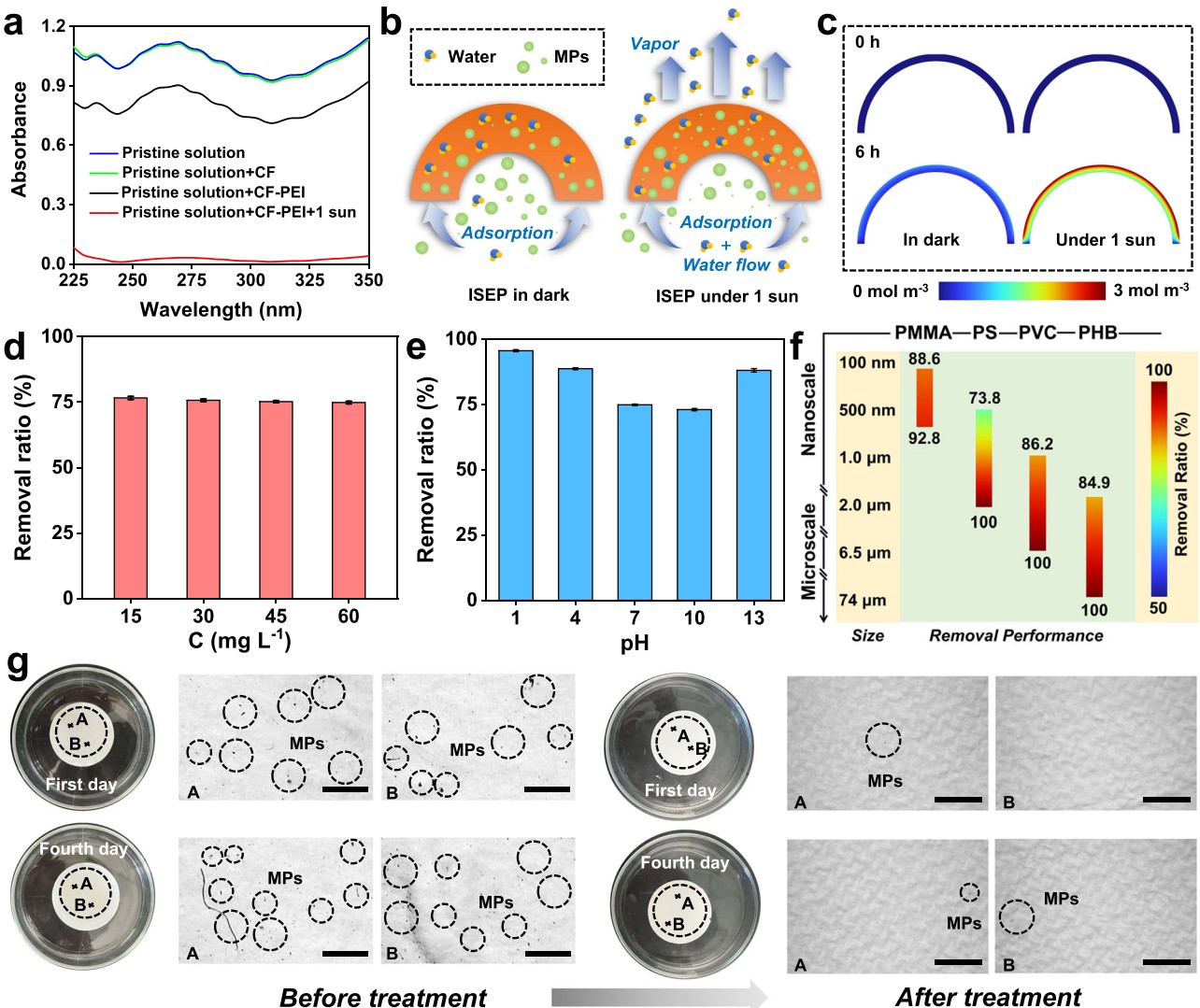

**Fig. 4 | The MPs removal mechanism and performance of ISEP. (a)** UV/Vis spectra of pristine PS solution, PS solution treated by ISEP based on CF and CF-PEI in dark, and PS solution treated by ISEP based on CF-PEI under 1 sun; **(b)** The mechanism for removing MPs by ISEP under 1 sun; **(c)** The MPs distribution in the CF-PEI of ISEP in dark or under 1 sun simulated by COMSOL; The MPs removal ratio by ISEP under various conditions: **(d)** Different MPs concentrations. Initial conditions: the pH of MPs solution is 7, and the particle size of MPs is 500 nm. Error bars represent the standard deviations from three-time measurements; **(e)** Different pH of MPs solution. Initial conditions: MPs concentration is 15 mg L$^{-1}$, and the particle size of MPs is 500 nm. Error bars represent the standard deviations from three-time measurements; **(f)** Different types of MPs with particle sizes ranging from the nanoscale to the microscale. Initial conditions: MPs concentration is 15 mg L$^{-1}$, and the pH of MPs solution is 7. **(g)** Microscopic images of the filter membrane after filtering original seawater and seawater treated by ISEP. Scale bar: 100 μm.

thereby achieving higher MPs enrichment performance. COMSOL simulations validate this hypothesis effectively. Compared to conditions in darkness, the MPs enrichment performance was significantly enhanced under solar irradiation (Fig. 4c). Additionally, MPs enrichment performance correlated with solar flux, in which higher solar flux will result in a higher MPs enrichment performance (Supplementary Fig. 22). For the low solar flux and low-temperature conditions, introducing a waste heat or electrical heat module to enhance the MP enrichment performance should also be considered.

Indoor experiments were conducted to investigate the MPs removal performance of ISEP in different conditions. Unless stated otherwise, PS removal experiments were all completed by ISEP based on CF-PEI within 3 h under 1 sun. The MPs concentration showed no significant impact on the MPs removal ability of ISEP (Fig. 4d). Conversely, the pH of the MPs solution had a substantial effect on the MPs removal ability of ISEP (Fig. 4e). Specifically when the pH transitioned from 1 to 13, the MPs removal ratio initially decreased from 100% to 73.8% and then increased to 82%. These fluctuations in MPs removal

performance may be attributed to alterations in the surface potential of CF-PEI. Next, we examined the effect of the particle size of MPs. PS with the particle size of 500 nm, 1 μm, 2 μm, and mixed sizes were selected for this experiment (Supplementary Fig. 23). ISEP could remove 100% of MPs with 2 μm but only 73.8% for MPs with 500 nm (Supplementary Fig. 24). It is worth noting that with the removal time extended from 3 h to 6 h, ISEP can remove 100% of PS even if the pH of MPs solution is 7 and the size of MPs is 500 nm (Supplementary Fig. 25). In addition, the PS beads adsorbed on the fibers of CF-PEI remain in their original shape under light conditions (Supplementary Fig. 26), indicating that the PS beads enriched by ISEP will not cause new pollution. The effect of these miscellaneous ions on the MPs removal was also investigated, showing little impact on the MPs removal performance (Supplementary Fig. 27). Subsequently, the removal performance by 3D-ISEP on different types of MPs was examined. MPs with different types and shapes (Supplementary Fig. 23 and Supplementary Fig. 28), including polymethyl methacrylate (PMMA), PS, polyvinyl chloride (PVC), and PHB, with particle sizes

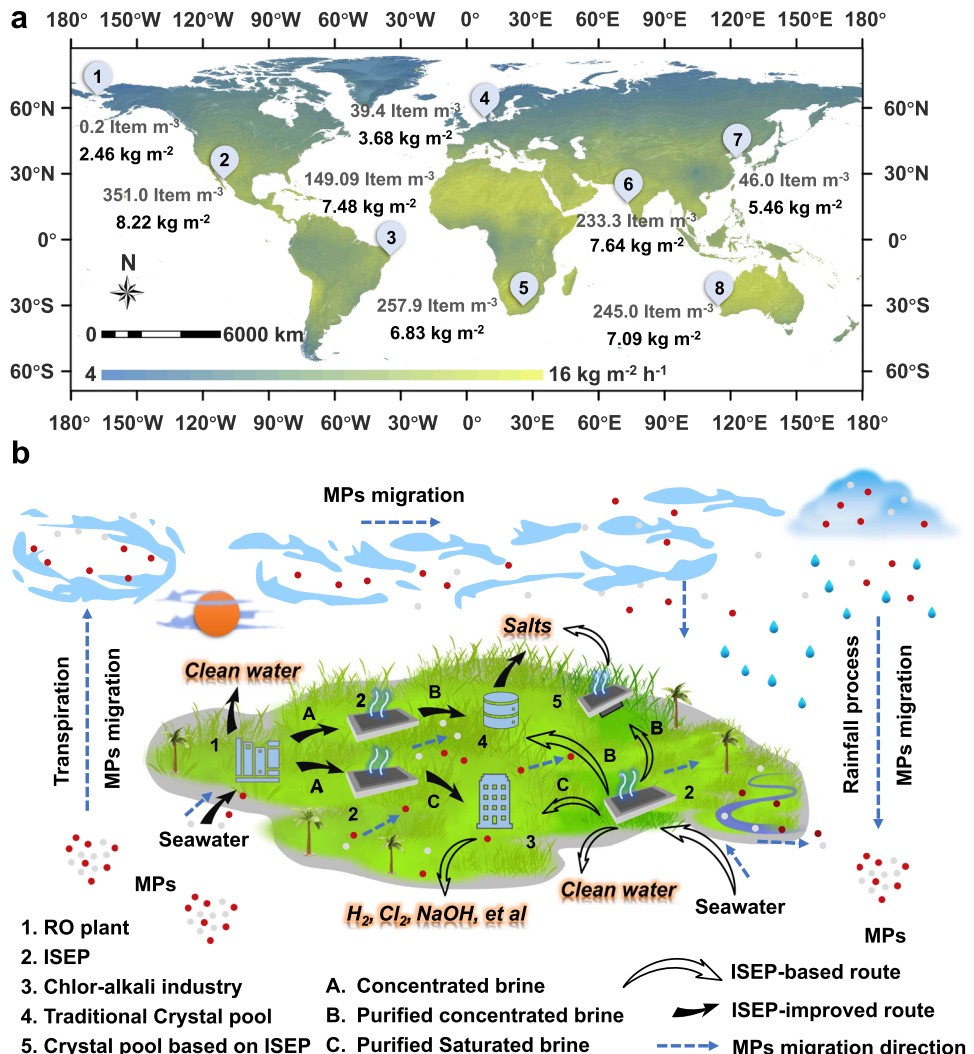

**Fig. 5 | The application for ISEP in the future. a** The map for the predicted daily outdoor evaporation rate worldwide. Inset: The pristine content of MPs (unit: Item m$^{-3}$) and the water production rate of the ISEP (Unit: kg m$^{-2}$) in the 8 representative areas[46–53]. **b** The prospective applications of ISEP.

ranging from the nanoscale to the microscale, were selected to conduct this experiment. The MPs removal performance of ISEP varied from 73.8% to 100% (Fig. 4f). MPs often aggregate under the action of salts, which means that small-size MPs will aggregate to form larger ones in seawater. We found that ISEP performs a higher MPs removal performance on large-size MPs, resulting in a higher MPs removal ratio in simulated seawater (Supplementary Fig. 29). It is also worth noting that the surface of PHB exhibits positive, and the surface of CF-PEI also shows positive (Supplementary Table 2). ISEP can still achieve a remarkable MPs removal ratio of 100%, which means that the MPs removal performance of 3D-ISEP is independent of the surface charge of the MPs.

Finally, outdoor experiments were conducted to investigate the practical MPs removal performance of 3D-ISEP. Numerous MPs were present in the original seawater, but these MPs were essentially undetectable after treatment with ISEP (Fig. 4g). The MPs removal rate exceeded 98%. The above results clearly demonstrate that our rationally-designed 3D-ISEP was highly effective in removing various MPs under complex conditions.

## Discussion

Two issues require further discussion, one of which pertains to clean water production. Despite claims by many studies that ISEP can produce clean water from practical seawater, our study's results suggest

these claims may be invalid due to the presence of MPs in the condensed water, which is concerning. It should be noted that not all clean water produced by ISEP contains MPs. Similar to ours (CF-PEI), ISEP prepared with MPs adsorbent can inhibit the enrichment of MPs in distilled water, but needs further study. Therefore, except for TOC and ion concentrations, MPs and other novel pollutants should also be examined to ensure whether the as-prepared condensed water is safe for drinking. We have estimated the evaporation rate of ISEP in various regions worldwide and selected 8 representative areas containing MPs for analyzing the water production potential of ISEP (Fig. 5a). ISEP installations near the Bering Strait (Point 1) and All Saints Bay (Point 2) exhibit the lowest and highest daily water production rate, approximately 2.46 and 8.22 kg m$^{-2}$, respectively. This output could meet the daily drinking needs of 1~3 people.

Some reported ISEPs seem to address the MPs pollution through collaborative photocatalytic processes[61,62]. Notably, these works focus on the simultaneous upgrading of MPs, while ours focuses on inhibiting the MPs accumulation in distilled water through collaborative physical processes. Although photocatalytic technology shows great potential in upgrading MPs, the following challenges may exist for ISEP coupled with photocatalytic technology in the real scenes: 1) Some smaller and less-stable MPs may inevitably form during the photocatalytic degradation, thereby bringing novel pollution[63,64]; 2) Given that the outdoor changeable solar flux and complex ingredients

(contains various MPs), multiple products may be formed, which presents challenges for product separation, especially clean water[62,65]; 3) The low light intensity in the real scenes may seriously limit the photocatalytic process, but the evaporation rate of ISEP is still high, thus resulting in the condensed water may also contain MPs. In contrast, the ISEP with a coordinated adsorption process is more universal in actual environments. We thus believe the photocatalytic process may be more suitable for treating adsorption-saturated ISEP.

Another issue concerns MPs removal: Initially, we only used CF-PEI as a proof of concept to investigate the enhanced MPs removal of ISEP. We also synthesize ISEPs using various reported MPs adsorbents. As expected, exposure to solar irradiation significantly improved the MPs removal performance, with the removal ratio increasing by up to 5.5 times (Supplementary Table 3). It is essential to emphasize that the enhancement effect of ISEP for various adsorbents on MPs removal ability differs, probably depending on the adsorption and evaporation performance of the adsorbent itself.

After adsorbed saturated, the MPs can be treated by various existing processes (such as photocatalysis, enzyme catalysis, thermal catalysis, etc.)[61,62,66,67]. Herein, we take the rapid joule heating technology (RJH) as an example to treat PS in the CF-PEI. Compared with traditional thermal treatment technology, RJH presents a low energy consumption[68]. The upgrading process of PS can be achieved within a few seconds (Supplementary Fig. 30). As a result, the PS in CF-PEI is mainly converted into $H_2$, with a high conversion ratio exceeding 90% (Supplementary Table 4). As for the treated CF-PEI, it can be used to absorb MPs again by growing PEI. However, these aspects fall beyond the scope of this work and warrant further study.

In addition to condensed water, certain seawater industrial products, such as sea salt or sodium hydroxide (Supplementary Fig. 31a and Supplementary Fig. 32a), may also contain MPs. To address this issue, we propose two innovative approaches (Fig. 5b): (1) Integration of ISEP with traditional seawater industries. Specifically, ISEP can serve as a pretreatment process to eliminate MPs from concentrated seawater beforehand. In this case, the sea salt and sodium hydroxide do not contain MPs (Supplementary Fig. 31b and Supplementary Fig. 32b). (2) ISEP can serve as an independent technology to achieve seawater desalination and simultaneously produce sea salt. However, further research is needed to assess the economics and feasibility of these two approaches.

Finally, the removal of MPs from the natural water environment could significantly reduce the presence of MPs in the biosphere. As part of the Earth's material cycle, certain MPs present in both the water environment and soil can enter the atmosphere through transpiration, while MPs in the atmosphere ultimately return to the water environment and soil through rainfall (Fig. 5b)[51]. Therefore, continuously reducing MPs in the water environment shows the great potential to significantly minimize their content in the biosphere.

## Methods
### Preparation of CF-PEI
CF-PEI was prepared by in situ grafting method[17,69]. Typically, CF was ultrasonically cleaned with absolute ethanol, acetone, and deionized water. The dried and cleaned CF with an area of 4 cm × 10 cm was immersed in PEI solution (5 mg·mL⁻¹) for 3 h at room temperature, and then glutaraldehyde (5 mg·mL⁻¹) was added. After being stirred for 2 h at room temperature, the as-obtained samples were washed with deionized water several times and vacuum-dried in an oven at 80°C for 12 h. After repeating the above operation 3 times, CF-PEI was obtained.

### Indoor solar evaporation experiments
The structure of the solar evaporator containing the CF and CF-PEI was shown in Fig. 3a. The solar evaporation experiment was conducted on a homemade optical system[37,56,70,71]. The evaporation rate was calculated by the mass change of the evaporation system before and after solar irradiation. The evaporation performance measurement method in the solution containing MPs was similar to that in pure water. The long-term evaporation experiments were conducted in real seawater (from the East China Sea). The seawater was treated by vacuum filtration (the pore size of the filter membrane is 80 μm). The obtained filtrate was used for further experiments. 3D-ISEP operated in seawater under 1 sun for 10 h and then transferred to the pristine solution for 14 h in the dark, forming a cycle. Unless mentioned, the evaporation performance measurement method in seawater was similar to that in pure water. A homemade device with a semicircular top was used to collect the condensed water produced by ISEP (Supplementary Fig. 8).

### Indoor MPs removal experiments
Firstly, the MPs removal experiments were performed at room temperature using the ISEP based on CF and CF-PEI at a neutral pH (6.5–7.0) under 1 sun or in the dark, taking the fluorescent PS as an example. Then the MPs removal experiments were conducted in various solutions with different MPs sizes, MPs concentrations, and pH values. The pH of the solution was adjusted by hydrochloric acid and sodium hydroxide. The removal experiments were carried out for 3 h and then analyzed using UV-Vis spectroscopy to determine the concentration of the residual MPs in the condensed water. Furthermore, the removal ratio ($\varphi$) was evaluated using Eq. (1) where $C_0$ was the initial concentration (mg L⁻¹) and $C_t$ was the equilibrium concentration (mg L⁻¹) of MPs in the solution, respectively.

$$\varphi = \frac{C_0 - C_t}{C_0} \tag{1}$$

The measurement method for the MPs removal performance of ISEP based on CF-PEI in MPs solutions with different particle sizes and types was similar to the above.

### Reporting summary
Further information on research design is available in the Nature Portfolio Reporting Summary linked to this article.

## Data availability
The data that supports the findings of the study are included in the main text and supplementary information files. Raw data can be obtained from the corresponding author upon request.

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

## Acknowledgements

We thank Prof. G. Chen's group and Prof. S. Cheng's group for advice on the whole project. We acknowledge Dr. C. Sun provided the technical support and assistance in rapid joule heating technology. We also acknowledge Dr. Z. Yu provided the water samples of the Qinghai-Tibet Plateau. This work was financially supported by National Natural Science Foundation of China (No. 42377380, X.-H.L.), National Key Research and Development Program of China (2019YFC1407800, X.-H.L.), China Postdoctoral Science Foundation (No. 2023M742588, Z.Y.) and Singapore Ministry of Education (A-8002144-00-00, S.-W. T.).

## Author contributions

X. L., S. T. and P. X. conceived and planned this research. Z.Y., Y. L. and B.M. did the experiments. C. L. and W. L. contributed to the theoretical analysis. Z.Y. and Y. Z. organized the data and wrote the manuscript. All authors discussed the results and approved the final version of the manuscript.

## Competing interests

The authors declare no competing interests.
