## [Peer Review File · Nature Communications]

Microplastic Detection and Remediation through Efficient Interfacial Solar Evaporation for Immaculate Water ProductionREVIEWER COMMENTS

Reviewer #1 (Remarks to the Author):

The authors proposed a new method for the removal of micropalstics and water purification by an interfacial solar evaporation platform (ISEP) inspired from the transpiration of plants. Although the idea is interesting, there are some limitations in this ISEP. The reviewer found the conclusion can not be well supported by the data at the present form.

1)The removal of micropalstics by solar energy without catalysis has been demonstrated recently, thus a comparison in the introduction section is necessary to justify the novelty of the work: "Sustainable removal of nano/microplastics in water by solar energy. Chemical Engineering Journal, 2022, 428, 131196. <https://www.sciencedirect.com/science/article/abs/pii/S1385894721027777>".

2)The water is just evaporated in the ISEP without indicating how to collect the purified water. Therefore, it sounds quite conceptual to the reader. I didn't see how the stone hit the other bird, although the authors claimed that "one stone kills two birds".

3)The reviewer also found the difficulties of the ISEP in the real application. For instance, in the sea water, the high concentration salt could precipitate and block the pores in the evaporation process that significantly reduce the function of ISEP. Besides, there are also salts in drinking water, the issue seems unavoidable.

4)The capacity of the ISEP should be further considered. It seems that the ISEP can not be reused. How to deal with the ISEP after the saturation of micropastics should also be discussed.

Reviewer #2 (Remarks to the Author):

Comment: In this paper, a water purification platform material with dual effect of evaporation and adsorption was prepared by in-situ deposition of PEI on commercial CF, and the dual function of fresh water conversion and MPs removal was successfully realized. According to "Drinking boiled tap water reduces human intake of nanoplastics and microplastics," ultra-efficient water recovery technology could even be combined with the thermal industry in the future. In this work, authors used solar green energy to assist evaporation, which was in line with the concept of green environmental protection and sustainable development. Overall, this work was worthy of encouragement and recognition, and the article had clear charts, comfortable colors and elegance. Finally, I would like to ask the author several questions to satisfy my curiosity and encourage the author to make further improvements:

Response:

Comment 1: According to the property of materials in the paper, temperature is a key factor in the realization of fresh water evaporation and MPs capture. According to the "Autonomous atmospheric water seeping MOF matrix", the MOF matrix can rise from room temperature to 53 °C in just 5 minutes under solar radiation, while ISEP seems to change temperature too slowly. This may lead to its inability to function as an effective freshwater purification collector in harsh environments.

Comment 2: According to "Autonomous atmospheric water seeping MOF matrix", MOF matrix has excellent uninterrupted work and cycle performance in collecting water, does ISEP in this paper also have it?

Comment 3: According to "Autonomous atmospheric water seeping MOF matrix", does ISEP have the characteristic of not actively condensing at lower temperatures? In low temperature environment will lose efficient pollutant removal performance, is there a way to further improve and control it?

Reviewer #3 (Remarks to the Author):

This work shows a “one stone kills two birds” strategy for removing MPs and producing clean water simultaneously. The MPs removal ratio increases by up to 5.5 times compared to the previously reported MPs adsorbents. This work is outstanding and timely. It can be accepted after a major revision. The specific comments are as follows:

(1) The authors state that this work is the first report on simultaneous clean water harvesting and MPs removal by solar evaporation. However, there are previous reports demonstrating this idea (see ref PNAS, 121 (13), e2317192121, J. Mater. Chem. A, 2021,9, 11013, et al.). Please check the statements related to novelty.

(2) This strategy has no function in MPs degradation or conversion, so how to avoid the blocking of this material by MPs during rapid and long-term solar evaporation?

(3) Can the high salinity (>3.5 wt%) in seawater influence the MPs removal performance? The charges in salts can induce the aggregation of MPs and change the MPs' size, thus influencing the absorption, filtration, and interception process of MPs by the CF-PEI.

(4) We suggest authors try to recycle the CF-PEI filled with MPs after water treatment. The strategy in this work can concentrate MPs in CF-PEI. If it can be upcycled into value-added chemicals, fuels, or new polymers, the work will be more meaningful.

(5) Table 1 can be removed to supporting information.

Response to Reviewer #1:

The authors proposed a new method for the removal of micropalstics and water purification by an interfacial solar evaporation platform (ISEP) inspired from the transpiration of plants. Although the idea is interesting, there are some limitations in this ISEP. The reviewer found the conclusion can not be well supported by the data at the present form.

Our response: We are immensely grateful to the Reviewer's comments and suggestions, which have greatly contributed to enhancing the quality of our manuscript. It is truly exciting to hear that Reviewer #1 finds our work interesting. We have diligently conducted additional experiments and provided further discussions to address the concerns raised by the Reviewer. All changes have been highlighted in red color in the revised manuscript. We hope that the revised manuscript will meet the expectations of Reviewer #1 and be suitable for publication in *Nature Communications*.

1. The removal of micropalstics by solar energy without catalysis has been demonstrated recently, thus a comparison in the introduction section is necessary to justify the novelty of the work: "Sustainable removal of nano/microplastics in water by solar energy. Chemical Engineering Journal, 2022, 428, 131196. <https://www.sciencedirect.com/science/article/abs/pii/S1385894721027777>".

Our response: Thanks for the Reviewer's comment. We would like to provide our elaborations further here to address the Reviewer's concern:

1) We have made a groundbreaking discovery the presence of MPs in distilled water produced by ISEP for the first time, a phenomenon not reported in the above references. This revelation undermines the assumption that distilled water is inherently clean in this scenario. We believe that this finding is undoubtedly disruptive and warrants significant attention.

2) In response, we innovatively combined the adsorption process with the solar-driven interfacial evaporation process, enabling good MPs interception effects. Our well-designed ISEP generated condensed water free from MPs under the effect of the

MPs adsorbent, a feature absent in traditional solar evaporation structures

More importantly, our work focuses on producing clean water free of MPs through interfacial solar evaporation technology, while the above-mentioned literature focuses on developing a novel MPs removal technology driven by solar energy, which is similar to the traditional air flotation. Although compared with traditional air flotation, the solar-driven microbubble technology enables lower energy consumption and operation costs, but electrostatic repulsion between microbubbles and MPs with negative charges probably makes it difficult to collect the MPs with negative charges (Water Environment Research 93.5 (2021): 693-702). Meanwhile, this microbubble technology could not produce clean water from real seawater or wastewater given that there are other pollutants or salts. In contrast, the ISEP with a coordinated adsorption process is more universal in actual environments to boost MPs removal and simultaneously produce clean water, which is extremely superior.

To solve the Reviewer's concern, we discussed the above references in detail in the revised version. The following changes have been made:

Page 2:

Some physical technologies have emerged as effective solutions for treating MPs¹⁷. For instance, Fang et al. recovered MPs through microbubbles produced by focusing sunlight through a glass ball¹⁸. While this technology offers lower energy consumption and operational costs than traditional air flotation technology, the electrostatic repulsion between microbubbles and negatively charged MPs probably poses a challenge for efficient collection¹⁹. Moreover, this approach is limited in producing clean water from real seawater or wastewater due to the presence of other pollutants or salts.

Reference:

18. Wang P, *et al.* Sustainable removal of nano/microplastics in water by solar energy. *Chemical Engineering Journal* **428**, 131196 (2022).

2. *The water is just evaporated in the ISEP without indicating how to collect the purified water. Therefore, it sounds quite conceptual to the reader. I didn't see how the stone hit the other bird, although the authors claimed that "one stone kills two birds".*

Our response: Thanks for the Reviewer’s comment. Our well-designed ISEP is fabricated by the MPs absorbents. Such ISEP generated condensed water free from MPs under the effect of the MPs adsorbent, a feature absent in traditional solar evaporation structures (**Figure R1**). Meanwhile, a homemade device with a semicircular top is used to collect the condensed water produced by such ISEP (**Figure R2**). During the initial 6 hours, a significant amount of water mist formed on the top of the device and gradually accumulated at the bottom due to gravity. Condensed water could then be collected from the bottom of this device for further analysis. The outdoor device is similar to the indoor device. By this way, ISEP could achieve “one stone kills two birds” that can produce clean water and simultaneously boost MPs removal.

Figure R1. Number of MPs (PS, mixed size, negatively charged) in different water samples determined by flow cytometry.

Figure R2. The schematic and digital photos of the homemade device for collecting the distilled water produced by ISEP.

To make it clear, we have added the relevant experiments and further elaboration in the revised version. The following changes have been made:

Page 3:

It is noteworthy that the ISEP generated condensed water free from MPs under the effect of the MPs adsorbent, a feature absent in traditional solar evaporation structures.

Page 6:

A homemade device with a semicircular top is used to collect the condensed water produced by ISEP (**Supplementary Fig. 8**).

Page 6:

The outdoor device is similar to the indoor device. Note that ISEP does not have an active condensation module and mainly relies on gravity to collect condensed water. Therefore, when the outdoor temperature is higher than the freezing point ($\sim 4^{\circ}\text{C}$), condensed water can be obtained, which endows it with a huge practical application potential.

Page 10:

It should be noted that not all clean water produced by ISEP contains MPs. Similar to ours (CF-PEI), ISEP prepared with MPs adsorbent can inhibit the enrichment of MPs in distilled water, but needs further study.

Supplementary Fig. 8:

Supplementary Fig.8. The schematic and digital photos of the homemade device for collecting the distilled water produced by ISEP. During the initial 6 hours, a significant amount of water mist formed on the top of the device and gradually accumulated at the bottom due to gravity. Condensed water could then be collected from the bottom of this device for further analysis.

3. The reviewer also found the difficulties of the ISEP in the real application. For instance, in the sea water, the high concentration salt could precipitate and block the pores in the evaporation process that significantly reduce the function of ISEP. Besides,

there are also salts in drinking water; the issue seems unavoidable.

Our response: We appreciate the Reviewer's comment and would like to provide further elaborations to address the reviewer's concern:

1) The outdoor desalination process of 3D-ISEP operates in an intermittent mode (Nano-Micro Letters 15.1 (2023): 214; Adv. Energy Mater. 2019, 9, 1900310; Chemical Engineering Journal 401 (2020): 126108). In this mode, the salt ions accumulating in the 3D-ISEP during the daytime will diffuse back into the seawater at night due to the concentration differences. The aforementioned literature has shown that for brine with a salinity lower than 10 wt.%, most ISEP can operate stably in this mode without salt precipitation on their surfaces (Separation and Purification Technology 301 (2022): 121938).

2) To further address the Reviewers' concerns, we also conducted long-term solar desalination experiments in actual seawater with the intermittent operation mode. In this mode, 3D-ISEP operated in seawater under 1 sun for 10 h and then transferred to the original seawater for 14 h in darkness, forming a cycle. During the ongoing 70-hour test, the 3D-ISEP maintained an evaporation rate of $\sim 1.95 \text{ kg m}^{-2} \text{ h}^{-1}$, showcasing its stable solar-powered desalination performance. Therefore, ISEP would not suffer from salt precipitation issue in practical applications.

3) Furthermore, similar to other ISEPs, the distilled water produced by 3D-ISEP contains negligible salts, well below the drinking water standards of WHO and EPA (**Figure R3**).

Figure R3. The ion concentrations of the original seawater and condensed water produced by 3D-ISEP.

In the revised version, we have added relevant experiments and further elaboration. The following changes have been made:

Page 7:

Except for water quality, stability is another factor to determine the practical potential of ISEP. The outdoor desalination process of ISEP belongs to an intermittent operation mode in a real scene^{56,57}. In this mode, the salt ions accumulating in the ISEP during the daytime will diffuse back into the bulk seawater at night due to the concentration differences, thereby achieving stable desalination^{58, 59, 60}. To verify it, we conduct long-term solar desalination experiments in actual seawater with this mode. During the ongoing 70-hour test, ISEP maintains an evaporation rate of $\sim 1.95 \text{ kg m}^{-2} \text{ h}^{-1}$ and no MPs are observed in the condensed water (**Supplementary Fig. 16**), demonstrating the stable solar-powered desalination performance of ISEP.

Supplementary Fig. 16:

Supplementary Fig. 16. The evaporation rate of 3D-ISEP based on CF-PEI in the long-term desalination experiments under 1 sun.

4. The capacity of the ISEP should be further considered. It seems that the ISEP can not be reused. How to deal with the ISEP after the saturation of micropastics should also be discussed.

Our response: Thanks for the Reviewer's comment. We would like to provide further elaborations to address the Reviewer's concern: 1) To determine the adsorption limit value, we conducted experiments to measure the water production and purification performance of ISEP in the MPs solution. The condensed water was collected every 6 h and analysed. Over a continuous 60-hour period, the 3D-ISEP demonstrated a stable average water collection rate of $1.21 \text{ kg m}^{-2} \text{ h}^{-1}$ with minimal detection of MPs in the

condensed water. With the operation time extending to 66 h, the average water production rate has little change but some MPs were detected in the condensed water, this may be related to the adsorption saturation of CF-PEI on MPs. In this state, to ensure the interception effect of MPs, it is necessary to regenerate CF-PEI. Note that the concentration of MPs in the real ocean is much lower than the above MPs solution ($0.2 \sim 351$ vs 69687 item m^{-3}). Under this condition, it is estimated that CF-PEI needs to be regenerated after being used at least 496 days (~ 198 times that in the indoor experiment). Given the lower light intensity outdoors, regeneration time may take longer. Similar to RO membranes, this regenerate frequency is acceptable. (2) The ISEP based on CF-PEI after adsorption saturation of MPs could be treated by various existing processes (such as photocatalysis, enzyme catalysis, thermal catalysis, etc). Herein, we take the rapid joule heat technology (RJH) as an example to treat PS in the CF-PEI. Compared with traditional thermal treatment technology, RJH presents a low energy consumption (One Earth 5, 1394-1403 (2022)). The upgrading process of PS can be achieved within a few seconds. As a result, the PS in CF-PEI is mainly converted into H_2 , with a high conversion ratio exceeds 90%. As for the treated CF-PEI, it can be used to absorb MPs again by growing PEI.

To make it clear, we have added the relevant experiments and further elaboration in the revised version. The following changes have been made:

Page 6:

To determine the adsorption limit value, we measure the water production and purification performance of ISEP in the above PS solution. The condensed water is collected every 6 h and analyzed. In the continuous 60 h, 3D-ISEP presents a stable average water collection rate of $1.21 \text{ kg m}^{-2} \text{ h}^{-1}$ and little MPs could be detected in the condensed water (**Supplementary Fig. 9**). With the operation time enlarging to 66 h, the average water production rate has little change but some MPs appears in the condensed water, this may be related to the adsorption saturation of CF-PEI on MPs. In this state, to ensure the interception effect of MPs, it is necessary to regenerate CF-PEI, which will be discussed below. Note that the concentration of MPs in the real ocean is much lower than the above MPs solution ($0.2 \sim 351$ vs 69687 item m^{-3})^{46, 47, 48, 49, 50, 51, 52, 53}. Under this condition, it is

estimated that CF-PEI needs to be regenerated after being used at least 496 days (The related details are provided in **Supplementary Method**). Given the lower light intensity outdoors, regeneration time may take longer. Similar to RO membranes, this regenerate frequency is acceptable. Therefore, during long-term operation process, MPs would not block the ISEP before regeneration and deteriorate the evaporation performance.

Page 12:

After adsorbed saturation, the MPs can be treated by various existing processes (such as photocatalysis, enzyme catalysis, thermal catalysis, etc.)^{61, 62, 66, 67}. Herein, we take the rapid joule heating technology (RJH) as an example to treat PS in the CF-PEI. Compared with traditional thermal treatment technology, RJH presents a low energy consumption⁶⁸. The upgrading process of PS can be achieved within a few seconds (**Supplementary Fig. 30**). As a result, the PS in CF-PEI is mainly converted into H₂, with a high conversion ratio exceeding 90% (**Supplementary Table 4**). As for the treated CF-PEI, it can be used to absorb MPs again by growing PEI. However, these aspects fall beyond the scope of this work and warrant further study.

Page 3 in Supplementary Information:

1.6 The estimation for the regeneration time of ISEP in an outdoor scene

Before estimation, two issues need to be further clarified:

- (1) ISEP has the same adsorption capacity to MPs outdoors and indoors
- (2) This estimation process ignored the effect of environmental factors such as temperature and humidity.

The regeneration time of ISEP in an outdoor scene was calculated by **Eq. S1**:

$$T_o = \frac{C_i}{C_o} T_i \quad (\text{S1})$$

where T_o was the regeneration time of ISEP in an outdoor scene (h); T_i was the regeneration time of ISEP in an indoor scene (60 h); C_i was the concentration of MPs in the indoor experimentes (69687 item m⁻³); C_o was the concentration of MPs in the real seawater (0.2 ~ 351 item m⁻³).

1.7 Rapid joule heat treatment on adsorbed PS

The upcycling of PS in CF-PEI using the fast joule heating technique is conducted on the

classic sealed tube reactor in an inert atmosphere. After adsorbed saturation to PS, CF-PEI is dried in an oven and then connected to the electrical power source in the quartz tube reactor. Before the reaction, adequate N₂ is pumped through the reactor during the whole reaction. After pumping N₂ for at least 10 min, a current pulse of 60 A is applied to the CF-PEI to decompose the adsorbed PS in 4 s. The released gas was captured in a 1 L gas bag and analyzed by gas chromatography (GC, FULI INSTRUMENTS GC9790 Plus, China).

Supplementary Fig. 9:

Supplementary Fig. 9. The average water production rate (AWPR) and MPs interception ratio (MIR) in the continuous 66 h under 1 sun.

Supplementary Fig. 30:

Supplementary Fig. 30. The digital photos of the upcycle and conversion of PS absorbed by CF-PEI by fast flash joule heating technology.

Supplementary Table 4:

Supplementary Table 4. The gas product obtained by upgrading PS (absorbed by CF-

PEI) through fast joule heating technology

Gas composition	Conversion ratio (%)	Gas composition	Conversion ratio (%)
H₂	93.2 ± 0.2	CH₄	2.0 ± 0.1
C₂H₄	4.7 ± 0.1	C₂H₆	0.03 ± 0.001
C₃H₆	0.06 ± 0.003	C₃H₈	0.03 ± 0.004

Response to Reviewer #2:

Comment: In this paper, a water purification platform material with dual effect of evaporation and adsorption was prepared by in-situ deposition of PEI on commercial CF, and the dual function of fresh water conversion and MPs removal was successfully realized. According to "Drinking boiled tap water reduces human intake of nanoplastics and microplastics," ultra-efficient water recovery technology could even be combined with the thermal industry in the future. In this work, authors used solar green energy to assist evaporation, which was in line with the concept of green environmental protection and sustainable development. Overall, this work was worthy of encouragement and recognition, and the article had clear charts, comfortable colors and elegance. Finally, I would like to ask the author several questions to satisfy my curiosity and encourage the author to make further improvements:

Our response: We gratefully appreciate the Reviewer's careful reading and strong recommendation of our manuscript. The manuscript has been thoroughly revised according to the Reviewer's insightful suggestions, as detailed below. We have tried our best to conduct additional experiments and provide further discussions to address the reviewer's concerns. All changes have been highlighted in red in the revised manuscript. We hope that the revised manuscript is suitable for publication in *Nature Communication*.

1. According to the property of materials in the paper, temperature is a key factor in the realization of fresh water evaporation and MPs capture. According to the "Autonomous atmospheric water seeping MOF matrix", the MOF matrix can rise from room temperature to 53 °C in just 5 minutes under solar radiation, while ISEP seems to change temperature too slowly. This may lead to its inability to function as an effective freshwater purification collector in harsh environments.

Our response: Thanks for the Reviewer's insightful comment. We would like to further provide our elaborations here to address reviewer's concern: 1) Although the steady-state temperature of CF-PEI in the wetted state is lower compared to the MOF matrix, it still presents a high temperature-rising rate. Specifically, the surface

temperature of CF-PEI can rise from 20.0 °C to 36.1 °C within 5 minutes, and then stabilized at 37 °C (**Figure R1**). The reported literature shows that surface temperatures above 35° are sufficient for ISEP to rapidly generate water vapour (Nature Communications 15.1 (2024): 874; Nature Communications 13, 1 (2022): 4335). Meanwhile, the higher surface temperature of ISEP may cause a lower evaporation rate due to excessive heat loss. Therefore, many scholars use three-dimensional evaporation structures to reduce the average temperature of the ISEP, thereby obtaining environmental energy and realizing the "cold evaporation process" (Joule 2.7 (2018): 1331-1338). 2) According to this comment, we supplied the evaporation performance of ISEP under low light intensity (0.2 sun, 0.4 sun, 0.6 sun and 0.8 sun) in the revised version. 3D-ISEP still enabled a stable evaporation rate of 0.61 kg m⁻² h⁻¹ under 0.2 sun, *ca* 2.7 times that in the dark. For other harsh environments (such as changeable weather, low temperature, no light, etc.), introducing phase change materials or waste heat to strengthen the evaporation process is needed (Advanced Energy Materials 13, 45 (2023): 2302451; Advanced Materials 35, 29 (2023): 2211932). To make it clear, we have added the evaporation experiments with low light intensity and further elaboration in the revised manuscript. The following changes have been made:

Page 5:

We further track the evaporation performance under low solar fluxes. 3D-ISEP still enables a stable evaporation rate of 0.61 kg m⁻² h⁻¹ under 0.2 sun, *ca* 2.7 times that in the dark (**Supplementary Fig. 6**). These results suggest that 3D-ISEP exhibits excellent evaporation performance even under weak solar irradiation, making it suitable for real-world application. For other extreme environments (such as changeable weather, low temperature, no light, etc.), introducing phase change materials or waste heat to strengthen the evaporation process is needed^{44, 45}.

Supplementary Fig. 6:

Supplementary Fig. 6. The evaporation rate of 3D-ISEP in pure water under low solar fluxes.

2. According to "Autonomous atmospheric water seeping MOF matrix", MOF matrix has excellent uninterrupted work and cycle performance in collecting water; does ISEP in this paper also have it?

Our response: Thanks for the Reviewer's insightful comment. Similar to MOF matrix, ISEP also have the uninterrupted work and cycle performance in collecting water. A homemade device with a semicircular top was used to collect the condensed water produced by ISEP (**Figure R1**). We measured the water production and purification performance of 3D-ISEP in the MPs solution under 1 sun. In the continuous 60 h, 3D-ISEP presented a stable water collection rate of 1.21 kg m⁻² h⁻¹. Therefore, ISEP also have the cycle performance in collecting water.

Figure R1. The schematic and digital photos of the homemade device for collecting the distilled water produced by ISEP.

To make it clear, we have added the relevant experiments and further elaboration in the revised version. The following changes have been made:

Page 6:

A homemade device with a semicircular top was used to collect the condensed water produced by ISEP (**Supplementary Fig. 9**).

Page 6:

To determine the adsorption limit value, we measure the water production and purification performance of ISEP in the above PS solution. The condensed water is collected every 6 h and analysed. In the continuous 60 h, 3D-ISEP presents a stable average water collection rate of $1.21 \text{ kg m}^{-2} \text{ h}^{-1}$ and little MPs could be detected in the condensed water (**Supplementary Fig. 9**).

Supplementary Fig. 8:

Supplementary Fig. 8. The schematic and digital photos of the homemade device for collecting the condensed water produced by ISEP. During the initial 6 hours, a significant amount of water mist formed on the top of the device and gradually accumulated at the bottom due to gravity. Condensed water could then be collected from the bottom of this device for further analysis.

Supplementary Fig. 9:

Supplementary Fig. 9. The average water production rate (AWPR) and MPs interception ratio (MIR) in the continuous 66 h under 1 sun.

3. According to "Autonomous atmospheric water seeping MOF matrix", does ISEP have the characteristic of not actively condensing at lower temperatures? In low temperature

environment will lose efficient pollutant removal performance, is there a way to further improve and control it?

Our response: Thanks for the Reviewer's insightful comment. Similar to the above reference, ISEP did not have an active condensation module. The condensation of the vapour produced by ISEP mainly relied on gravity. A homemade device with a semicircular top was used to collect the condensed water produced by ISEP (**Figure R1**). A significant amount of water mist formed on the top of the device and gradually accumulated at the bottom due to gravity under solar irradiation. Condensed water could then be collected from the bottom of this device for further analysis. Therefore, when the temperature was higher than the freezing point ($\sim 4^{\circ}\text{C}$), the condensed water can be obtained. As for the temperature reduced below freezing point, it may be necessary to introduce a waste heat or electrical heat module to enhance the performance (Advanced Materials 35. 29 (2023): 2211932; Advanced Materials 29.38 (2017): 1702590), which should be studied in future.

To make it clear, we have added the relevant experiments and further elaboration in the revised version. The following changes have been made:

Page 6:

A homemade device with a semicircular top was used to collect the condensed water produced by ISEP (**Supplementary Fig. 8**).

Page 6:

The outdoor device is similar to the indoor device. Note that ISEP does not have an active condensation module and mainly relies on gravity to collect condensed water. Therefore, when the outdoor temperature is higher than the freezing point ($\sim 4^{\circ}\text{C}$), condensed water can be obtained, which endows it with a huge practical application potential.

Page 9:

For the low solar flux and low-temperature conditions, introducing a waste heat or electrical heat module to enhance the MP enrichment performance should also be considered.

Supplementary Fig. 8:

Supplementary Fig. 8. The schematic and digital photos of the homemade device for collecting the distilled water produced by ISEP. During the initial 6 hours, a significant amount of water mist formed on the top of the device and gradually accumulated at the bottom due to gravity. Condensed water could then be collected from the bottom of this device for further analysis.

Response to Reviewer #3:

Comment: This work shows a one stone kills two birds strategy for removing MPs and producing clean water simultaneously. The MPs removal ratio increases by up to 5.5 times compared to the previously reported MPs adsorbents. This work is outstanding and timely. It can be accepted after a major revision.

Our response: We extend our sincere appreciation to the Reviewer for his/her meticulous review and positive comments on our manuscript. We have diligently addressed the Reviewer's insightful suggestions through thorough revisions. We have tried our best to conduct additional experiments and provide further discussions to address the concerns of the Reviewer. All changes have been highlighted in red throughout the revised manuscript. We are hopeful that the revised manuscript now meets the Reviewer's expectation and is suitable for publication in *Nature Communications*.

1. The authors state that this work is the first report on simultaneous clean water harvesting and MPs removal by solar evaporation. However, there are previous reports demonstrating this idea (see ref PNAS, 121 (13), e2317192121, J. Mater. Chem. A, 2021,9, 11013, et al.). Please check the statements related to novelty.

Our response: Thanks for the Reviewer's comment. We would like to provide our elaborations further here to address the Reviewer's concern:

1) We have made a groundbreaking discovery the presence of MPs in distilled water produced by ISEP for the first time, a phenomenon not reported in the above references. This revelation undermines the assumption that distilled water is inherently clean in this scenario. We believe that this finding is undoubtedly disruptive and warrants significant attention.

2) In response, we innovatively combined the adsorption process with the solar-driven interfacial evaporation process, enabling good MPs interception effects. Our well-designed ISEP generated condensed water free from MPs under the effect of the MPs adsorbent, a feature absent in traditional solar evaporation structures.

More importantly, our work focuses on reducing the MPs concentration in distilled

water through collaborative physical processes, while the above-mentioned literature focuses on the simultaneous upgrading of MPs in bulk water through collaborative photocatalytic processes. Although photocatalytic technology shows great potential in upgrading MPs, the following challenges may exist for ISEP coupled with photocatalytic technology in the real scenes:

A) Potential secondary pollution. As mentioned in J. Mater. Chem. A, 2021,9, 11013, et al (Original text: “In this case, smaller and less-stable microplastics featured with larger contact/charged surfaces inevitably formed during the photo-degradation...”), some smaller and less-stable MPs may inevitably form during the photocatalytic degradation, thereby bringing novel pollution;

B) Product separation challenges, especially clean water. As mentioned in the PNAS, 121 (13), e2317192121 (Original text: “With a conversion rate of 46.1% h⁻¹, the PET can be converted into 20.3 μmol g⁻¹ h⁻¹ of 1,4-benzenedicarboxaldehyde, 208 μmol g⁻¹ h⁻¹ of 1,4-dimethylbenzene, 130.6 μmol g⁻¹ h⁻¹ of ethylene glycol, 22.6 μmol g⁻¹ h⁻¹ of acetic acid (SI Appendix, Fig. S45), and 120.4 μmol g⁻¹ h⁻¹ of CO₂ (SI Appendix, Fig. S46)...”), various MPs in the real seawater would result in multiple products, which presents challenges for product separation, especially clean water.

C) The above literature did not conduct MPs treatment experiment under actual outdoor conditions. The low light intensity in the real scenes may seriously limit the photocatalytic process, but the evaporation rate of ISEP is still high (0.61 kg m⁻² h⁻¹ under 0.2 sun), thus resulting in the condensed water may also contain MPs. In contrast, the ISEP with a coordinated adsorption process is more universal in actual environments.

Thus, from a scientific perspective, we believe the photocatalytic process may be more suitable for treating adsorption-saturated ISEP. To solve the Reviewer's concern, we discussed the above references in detail and corrected our statement in the version to emphasize the innovative nature of the work. The following changes have been made:

Page 1:

In this work, we propose a “one stone kills two birds” strategy, employing an interfacial solar evaporation platform (ISEP) combined with a MPs adsorbent **for the first time**.

Page 3:

It is noteworthy that the ISEP generated condensed water free from MPs under the effect of the MPs adsorbent, a feature absent in traditional solar evaporation structures.

Page 10:

Although many studies claim they can produce clean water from practical seawater by ISEP, these indications may be invalid. Our study results first show that MPs would exist in the condensed water, which is undoubtedly subversive. It should be noted that not all clean water produced by ISEP contains MPs. Similar to ours (CF-PEI), ISEP prepared with MPs adsorbent can inhibit the enrichment of MPs in distilled water. Therefore, except for TOC and ion concentrations, MPs and other novel pollutants should also be examined to ensure whether the as-prepared condensed water is safe for drinking.

Page 11:

Some reported ISEPs seem to address the MPs pollution through collaborative photocatalytic processes^{61, 62}. Notably, these works focus on the simultaneous upgrading of MPs, while ours focuses on inhibiting the MPs accumulation in distilled water through collaborative physical processes. Although photocatalytic technology shows great potential in upgrading MPs, the following challenges may exist for ISEP coupled with photocatalytic technology in the real scenes: 1) Some smaller and less-stable MPs may inevitably form during the photocatalytic degradation, thereby bringing novel pollution^{63, 64}; 2) Given that the outdoor changeable solar flux and complex ingredients (contains various MPs), multiple products may be formed, which presents challenges for product separation, especially clean water^{62, 65}; 3) The low light intensity in the real scenes may seriously limit the photocatalytic process, but the evaporation rate of ISEP is still high, thus resulting in the condensed water may also contain MPs. In contrast, the ISEP with a coordinated adsorption process is more universal in actual environments. We thus believe the photocatalytic process may be more suitable for treating adsorption-saturated ISEP.

Reference:

61. Meng X, Peng X, Xue J, Wei Y, Sun Y, Dai Y. A biomass-derived, all-day-round solar evaporation platform for harvesting clean water from microplastic pollution. *Journal of Materials Chemistry A* **9**, 11013-11024 (2021).

62. Meng X, *et al.* Integration of photothermal water evaporation with photocatalytic microplastics upcycling via nanofluidic thermal management. *Proceedings of the National Academy of Sciences* **121**, e2317192121 (2024).

2. This strategy has no function in MPs degradation or conversion, so how to avoid the blocking of this material by MPs during rapid and long-term solar evaporation?

Our response: Thanks for the Reviewer's insightful comment. MPs would not block the ISEP before regeneration and deteriorate the evaporation performance during the rapid and long-term solar evaporation experiments. To determine the adsorption limit value, we conducted experiments to measure the water production and purification performance of ISEP in the MPs solution. The condensed water was collected every 6 h and analyzed. Over a continuous 60-hour period, the 3D-ISEP demonstrated a stable average water collection rate of $1.21 \text{ kg m}^{-2} \text{ h}^{-1}$ with minimal detection of MPs in the condensed water. With the operation time extending to 66 h, the average water production rate has little change but some MPs were detected in the condensed water, this may be related to the adsorption saturation of CF-PEI on MPs. In this state, to ensure the interception effect of MPs, it is necessary to regenerate CF-PEI. Note that the concentration of MPs in the real ocean is much lower than the above MPs solution ($0.2 \sim 351$ vs $69687 \text{ item m}^{-3}$). Under this condition, it is estimated that CF-PEI needs to be regenerated after being used at least 496 days (~ 198 times that in the indoor experiment). Given the lower light intensity outdoors, regeneration time may take longer. Similar to RO membranes, this regenerate frequency is acceptable. Therefore, during rapid and long-term solar evaporation, MPs would not block the ISEP before regeneration and deteriorate the evaporation performance. To make it clear, we have added the relevant experiments and further elaboration in the revised version. The following changes have been made:

Page 6:

To determine the adsorption limit value, we measure the water production and purification performance of ISEP in the above PS solution. The condensed water is collected every 6 h and analyzed. In the continuous 60 h, 3D-ISEP presents a stable average water collection

rate of $1.21 \text{ kg m}^{-2} \text{ h}^{-1}$ and little MPs could be detected in the condensed water (**Supplementary Fig. 9**). With the operation time enlarging to 66 h, the average water production rate has little change but some MPs appears in the condensed water, this may be related to the adsorption saturation of CF-PEI on MPs. In this state, to ensure the interception effect of MPs, it is necessary to regenerate CF-PEI, which will be discussed below. Note that the concentration of MPs in the real ocean is much lower than the above MPs solution ($0.2 \sim 351$ vs $69687 \text{ item m}^{-3}$)^{46, 47, 48, 49, 50, 51, 52, 53}. Under this condition, it is estimated that CF-PEI needs to be regenerated after being used at least 496 days (The related details are provided in **Supplementary Method**). Given the lower light intensity outdoors, regeneration time may take longer. Similar to RO membranes, this regenerate frequency is acceptable. Therefore, during long-term operation process, MPs would not block the ISEP before regeneration and deteriorate the evaporation performance.

Supplementary Fig. 9:

Supplementary Fig. 9. The average water production rate (AWPR) and MPs interception ratio (MIR) in the continuous 66 h under 1 sun.

3. Can the high salinity (>3.5 wt%) in seawater influence the MPs removal performance? The charges in salts can induce the aggregation of MPs and change the MPs'size, thus influencing the absorption, filtration, and interception process of MPs by the CF-PEI.

Our response: Thanks for the Reviewer's insightful comment. We are very greatly with Reviewer that the salts would induce the aggregation of MPs, but we thought it is beneficial for MPs removal. We would like to provide our elaborations further here to

address the Reviewer's concern: (1) The MPs we treated here were mainly small-size MPs (< 65 μm). Large-size MPs aggregated under the action of salt will be directly removed during the pretreatment process (filter), while small-size MPs will aggregate to form large ones. We have found that ISEP performed a higher MPs removal performance on large-size MPs. Therefore, we thought the high salinity (>3.5 wt%) in seawater would benefit the MPs removal performance. (2) We also measured the MPs removal performance of ISEP in the simulated seawater. ISEP enabled a higher MPs removal ratio in the seawater. Therefore, we thought that the high salinity (>3.5 wt%) in seawater would not reduce the MPs removal performance. To make it clear, we have added the relevant experiments and further elaboration in the revised version. The following changes have been made:

Page 10:

MPs often aggregate under the action of salts, which means that small-size MPs will aggregate to form large ones in seawater. We have found that ISEP performs a higher MPs removal performance on large-size MPs, resulting in a higher MPs removal ratio in simulated seawater (**Supplementary Fig. 29**).

Page 13:

The seawater was treated by vacuum filtration (the pore size of the filter membrane is 80 μm). The obtained filtrate was used for further experiments.

Supplementary Fig. 29:

Supplementary Fig. 29. The removal ratio of PS by ISEP in the PS solution prepared by pure water and simulated seawater. Initial conditions: the pH of MPs solution is 7, MPs concentration is 15 mg L^{-1} . The simulated seawater is prepared by a typical

procedure¹².

4. We suggest authors try to recycle the CF-PEI filled with MPs after water treatment. The strategy in this work can concentrate MPs in CF-PEI. If it can be upcycled into value-added chemicals, fuels, or new polymers, the work will be more meaningful.

Our response: Thanks for the Reviewer's insightful comment. The ISEP based on CF-PEI after adsorption saturation of MPs could be treated by various existing processes (such as photocatalysis, enzyme catalysis, thermal catalysis, etc). Herein, we take the rapid joule heat technology (RJH) as an example to treat PS in the CF-PEI. Compared with traditional thermal treatment technology, RJH presents a low energy consumption (One Earth 5, 1394-1403 (2022)). The upgrading process of PS can be achieved within a few seconds (**Figure R1**). As a result, the PS in CF-PEI is mainly converted into H₂, with a high conversion ratio exceeds 90%. As for the treated CF-PEI, it can be used to absorb MPs again by growing PEI.

Figure R1. The digital photos of the upcycle and conversion of PS absorbed by CF-PEI by fast flash joule heating technology.

To make it clear, we have added the relevant experiments and further elaboration in the revised version. The following changes have been made:

Page 12:

After adsorbed saturation, the MPs can be treated by various existing processes (such as photocatalysis, enzyme catalysis, thermal catalysis, etc.)^{61, 62, 66, 67}. Herein, we take the rapid

joule heating technology (RJH) as an example to treat PS in the CF-PEI. Compared with traditional thermal treatment technology, RJH presents a low energy consumption⁶⁸. The upgrading process of PS can be achieved within a few seconds (**Supplementary Fig. 30**). As a result, the PS in CF-PEI is mainly converted into H₂, with a high conversion ratio exceeding 90% (**Supplementary Table 4**). As for the treated CF-PEI, it can be used to absorb MPs again by growing PEI. However, these aspects fall beyond the scope of this work and warrant further study.

Page 3 in Supplementary Information:

1.7 Rapid joule heat treatment on adsorbed PS

The upcycling of PS in CF-PEI using the fast joule heating technique is conducted on the classic sealed tube reactor in an inert atmosphere. After adsorbed saturation to PS, CF-PEI is dried in an oven and then connected to the electrical power source in the quartz tube reactor. Before the reaction, adequate N₂ is pumped through the reactor during the whole reaction. After pumping N₂ for at least 10 min, a current pulse of 60 A is applied to the CF-PEI to decompose the adsorbed PS in 4 s. The released gas was captured in a 1 L gas bag and analyzed by gas chromatography (GC, FULI INSTRUMENTS GC9790 Plus, China).

Supplementary Fig. 30:

Supplementary Fig. 30. The digital photos of the upcycle and conversion of PS absorbed by CF-PEI by fast flash joule heating technology.

Supplementary Table 4:

Supplementary Table 4. The gas product obtained by upgrading PS (absorbed by CF-PEI) through fast joule heating technology

Gas composition	Conversion ratio (%)	Gas composition	Conversion ratio (%)
H₂	93.2 ± 0.2	CH₄	2.0 ± 0.1
C₂H₄	4.7 ± 0.1	C₂H₆	0.03 ± 0.001
C₃H₆	0.06 ± 0.003	C₃H₈	0.03 ± 0.004

5. Table 1 can be removed to supporting information.

Our response: Thanks for the Reviewer's suggestion. Table 1 has been transferred to the *Supporting information* in the revised version.

Supplementary Table 3:

Supplementary Table 3. The MPs removal ratio of previously reported adsorbents, ISEP fabricated by these adsorbents in dark and under 1 sun

Adsorbents	Reported MPs removal ratio	MPs removal ratio by ISEP in dark	MPs removal ratio by ISEP under 1 sun
Magnetic nano-Fe₃O₄ (Metal-based)¹²	32 ± 1.2% (30 min)	27 ± 2.3% (30 min)	88 ± 2.1% (30 min)
Magnetic carbon nanotubes (Carbon-based)¹³	14 ± 2.8% (60 min)	11 ± 1.2% (60 min)	72 ± 1.5% (60 min)
Cu-Ni carbon material (Hybrid-based)¹⁴	40 ± 3.6% (120 min)	34 ± 2.8% (120 min)	90 ± 2.1% (120 min)

REVIEWERS' COMMENTS

Reviewer #1 (Remarks to the Author):

The author have addressed all my concerns.I recommend its publication in Nature Communications.

Reviewer #2 (Remarks to the Author):

The authors have well responded to the reviewers' questions and revised the manuscript according to the comments. The revised manuscript has reached the standard for publication.

Reviewer #3 (Remarks to the Author):

The authors have addressed all my concerns. It can be accepted in current form.